# Trend of Injury Severity and Road Traffic-Related Mortality in an Arab Middle Eastern Country: A 12-Year Retrospective Observational Study

**DOI:** 10.3390/healthcare13091045

**Published:** 2025-05-01

**Authors:** Tarik Abulkhair, Rafael Consunji, Ayman El-Menyar, Tongai F. Chichaya, Mohammad Asim, Hassan Al-Thani

**Affiliations:** 1Trauma Surgery Section, Hamad Medical Corporation, Doha 3050, Qatar; tabuelkkeer@hamad.qa (T.A.); rconsunji@hamad.qa (R.C.); masim1@hamad.qa (M.A.); althanih@hotmail.com (H.A.-T.); 2Weill Cornell Medicine, Doha 24144, Qatar; 3School of Health and Care, Coventry University, Coventry CV1 5FB, UK; chichayatf@gmail.com

**Keywords:** trauma, injury prevention, road traffic injuries, mortality, injury severity

## Abstract

**Background:** Road traffic injuries (RTIs) significantly contribute to disability and death in Qatar. This observational study aimed to explore RTI mortality and injury severity trends from 2011 to 2022. **Methods:** Data from the national trauma database were analyzed retrospectively for mortality rates, injury severity, and characteristics of the injured populations over the years (2011–2022). **Results:** RTIs represented around 61.3% (*n* = 12,644) of 20,642 trauma hospitalizations over 12 years. The aggregate RTI mortality rate decreased from 12 to 8 per 100,000 persons, with a mean patient age of 31.8 years. The sum of deaths was 2464, comprising 1022 (41%) in-hospital and 1442 (59%) out-of-hospital fatalities. Among in-hospital deaths, bike-related mortalities totaled 35 (3%), motorcycle-related mortalities 53 (5%), motor vehicle mortalities 561 (55%), and pedestrian mortalities 373 (36%). Based on the injury severity score (ISS), RTIs were divided into four categories, namely, mild (ISS: 1–9), moderate (ISS: 10–15), severe (ISS: 16–24), and fatal (ISS: 25–75). The ISS ranged from 12 to 14, while the median ranged from 10 to 12. The injury frequency showed that mild injuries comprised 40.6% (4545), moderate injuries 26.2% (2934 subjects), and severe 16.7% (1873 subjects). Profound injuries accounted for 13.3% (1490 subjects). Severe and fatal injuries combined dropped from 30% in 2011 to 25% by 2022. Inversely, moderate injuries increased from 24% to 30%, representing a downward trend of the injury severity. Motorcycle-related injuries rose from around 3% to 28% between 2011 and 2022. Motor vehicle and pedestrian injuries declined from about 67% to 54% and 27% to 15%, respectively. Winter, Autumn, Spring, and Summer accounted for 27%, 26%, 24%, and 23% of the total injuries (11,153), respectively. **Conclusions:** RTI in-hospital mortality and injury severity decreased over the study period. Injury prevention programs should target frequent injury seasons and high-risk populations, such as motorcyclists.

## 1. Introduction

Road traffic injuries (RTIs) remain a serious problem for public health worldwide. The World Health Organization (WHO) reported that RTIs are the eighth most common cause of mortality across all age groups worldwide, accounting for an estimated 1.35 million deaths yearly. In addition, the number of individuals who experience non-fatal injuries per annum ranges between 20 and 50 million, often resulting in disabilities [1]. From an economic perspective, RTIs account for a loss of about 3% of the Gross Domestic Product (GDP) in many countries [1], and Qatar is no exception. RTIs are the second primary cause of death, and the leading cause of fatalities and disabilities combined, making up 30% of disability-adjusted life years [2]. Additionally, this RTI burden accounts for 2.7 percent of the country’s GDP, as estimated by the Road Safety Studies Center [3].

There is a lack of evidence about the impact of road safety initiatives on RTI outcomes in Qatar. Yet, several studies have evaluated RTIs in part. The United Nations’ goal to reduce RTIs underscores the pressing need for studies on this topic. Qatar has undertaken various road safety initiatives, including injury epidemiology, collaboration with ministries, and injury prevention campaigns [4]. Furthermore, Qatar’s commitment to the global DoARS (Decade of Action for Road Safety) initiative since 2011 reflects the country’s vision for enhanced road safety. The DoARS initiative [5] is focused on reducing the number of deaths and serious injuries from RTIs by implementing a five-pillar strategy. This strategy includes enhancing road safety management, upgrading infrastructure, ensuring vehicle safety, promoting road user behavior, and improving the post-crash response.

RTIs significantly affect individuals’ well-being, finances and societal resources, covering road repairs, lost employee productivity, and straining medical services [6]. For instance, from 2008 to 2010, an annual average of 220 individuals lost their lives and more than 550 people sustained serious RTIs in Qatar [6]. Moreover, the RTI death rate dropped from around 13 to 7 per 100,000 individuals between 2010 and 2016 [7]. Despite the progress, efforts are still required to address this problem. On a global scale, the UN declared 2011 to 2020 as the decade in which global road traffic fatalities would be reduced by 50%. However, despite the global DoARS implementation, RTI fatalities increased from 1.24 million in 2010 to 1.35 million in 2016 [7]. Thus, the UN recently announced the second DoARS recommendations, aiming to reduce road traffic mortality by 50% by 2030 [8]. The National Road Safety Strategy 2013–2022 (NRSS) postulated the approaches needed to produce a significant reduction in road traffic fatalities and injuries in Qatar by 2022 [6].

The present study’s objectives were to describe and analyze the trends and patterns of road traffic deaths, injury severity, and characteristics of injured populations between 2011 and 2022. The current study addressed the gap in the literature by analyzing the RTI trends in terms of mortality, injury severity, mechanism of injury, seasonality, and characteristics of the injured. This research also evaluated the progress in reducing RTI mortality and injury severity against national goals. Hence, the findings can assist policymakers and health promotion professionals in creating suitable prevention programs for the most vulnerable populations.

## 2. Materials and Methods

### 2.1. Procedure

The authors applied a quantitative retrospective observational design. This study examined demographic data (age and sex), injury mechanisms, mortality rates, and injury severity based on the injury severity score (ISS), as summarized in Table 1 and Table 2.

The ISS is an anatomical scoring system with a sum score ranging from 0 to 75. Each anatomical injury in the six body regions (head and neck, face, chest, abdomen, extremities, and skin) is scored from one (lowest severity) to six (highest severity). Only the highest score from each region is considered. The scores of the three most critically injured regions are squared and then summed to determine the ISS. Higher scores mean increased severity [9]. The ISS is also a common and effective measure of injury severity and mortality prediction [10]. The ISS reflects the adequacy of injury prevention measures taken by the road user during crashes, such as seat belts and helmets. The shock index, defined as the ratio of heart rate to systolic blood pressure, serves as a quick tool to assess hemodynamic stability, particularly useful in trauma settings to identify patients at risk of deterioration [11].

Descriptive statistics were used to analyze variations in injury severity and deaths during the period from 1 January 2011 to 31 December 2022. Categorical data were presented as frequencies and proportions, while continuous data were reported as means, medians, and rates. For statistical significance testing, differences in categorical variables between two groups were assessed using Chi-square tests. Additionally, continuous data involving more than two groups were compared using one-way ANOVA for normally distributed variables. The significance of statistical differences was attributed to a two-tailed *p*-value of <0.05. The raw data were exported into a spreadsheet (Microsoft Excel Version 2208 (Build 15601.20578)) and then imported into statistical analysis software. All analyses were performed using the Statistical Package for Social Sciences version 21 (SPSS Inc., Chicago, IL, USA).

### 2.2. Data Collection

This study utilized data from Qatar National Trauma Registry (QNTR) and Qatari Planning and Statistics Authority (QPSA), the latter of which comprises publicly available data source. A trauma registry is a specialized tool used in healthcare settings. The QNTR represents a repository of detailed information about each trauma patient, including demographic data, injury characteristics, prehospital care, emergency department interventions, surgical procedures, in-hospital treatments, and outcomes. These data were typically collected from medical records, diagnostic reports, and other relevant sources.

The QNTR follows data validation standards from the American College of Surgeons. Therefore, the registry has a comprehensive data validation process. This process involves a systematic and regular evaluation and verification of the data to ensure completeness, accuracy, and consistency among data abstractors for the validity of its results [12]. On the other hand, the QPSA publishes the overall numbers of RTI deaths across the country monthly [13]. Since the QNTR only had information about the hospitalized populations, the researchers obtained the prehospital death numbers by subtracting in-hospital deaths from the total deaths published by the QPSA.

In this study, RTI was defined as the mechanism of injury according to the external causes of injury listed in the International Classification of Diseases, Ninth Revision (ICD-9), specifically within the code range E810 to E819. This study included all injured populations from road crashes reported during those 12 years (12,644). The records that had missing data were minimal. This window was selected because it overlaps with DoARS completion (2011–2020) and NRSS implementation (2013–2022), which introduced multiple road safety initiatives.

## 3. Results

### 3.1. Mortality Findings

Overall, the annual death rate resulting from RTIs decreased over the study period from 12 to 8 per 100,000 persons. An RTI-related death is considered an individual who died at the crash scene (prehospital) or during the initial hospital encounter (in-hospital) following RTI. The total number of deaths was 2463, comprising 1021 in-hospital and 1442 prehospital fatalities, marking 41% and 59%, respectively.

The in-hospital death rate showed a downward trend from 5 to 2 per 100,000 people between 2011 and 2022. The prehospital death rate dropped from 7 deaths/100,000 in 2011 to 3 in 2019 but subsequently increased in later years, reaching 6 by 2022. The disparity between these rates has oscillated annually. However, there was an increasing trend in prehospital fatalities between 2018 and 2022 (Figure 1).

### 3.2. RTIs by Injury Severity Score

There was a shift from severe injuries to less severe ones. Among 11,202 hospitalized injured people, ISS was documented for 10,842 patients (97%) and missing data accounted for 3%. The ISS average ranges from 12 to 14, while the median ranges from 10 to 12. Over the twelve years studied, mild injuries (ISS: 0–9) were the most prevalent category, comprising 41.9% (4545 subjects). Moderate injuries (ISS: 10–15) increased from 24% to 30%, representing 27.1% (2934 subjects) overall. Severe injuries (ISS 16–24) peaked at 20% in 2016 and 2018, then declined to 14% by 2022, making up 17.3% (1873 subjects). A similar decline was noted among those with ISSs of 25–75 (fatal injuries). The injuries in this category decreased from 13% in 2011 to 11% in 2020–2022, representing 13.7% (1490 subjects) of the sum. In summary, while minor injuries remained relatively stable, the severe and fatal injuries collectively dropped from 30% to 25%, contrasting with a rise in moderate injuries from 24% to 30% during the study period (Figure 2).

### 3.3. Mechanism of Injury Findings

RTI mechanisms showed remarkable changes during the study period. Of the hospitalized patients (11,202), fewer than 1% had missing injury mechanism data. Among 11,193 people, MVC represented the largest category (65%), followed by pedestrian injuries (PED) (22%), motorcycle injuries (MCC) (9%), and bicycle injuries (4%). Between 2011 and 2020, the annual average of bicycle injuries was around 3% (30). However, it rose to about 6% in 2021 and then returned to 3% the following year. MCC injuries ranged from 3 to 5% (2011–2016) but increased gradually from around 5% (43) in 2017 to 28% (326) by 2022. MVC-related injuries fluctuated around the mid-60s in percentages (2011–2021). The most noticeable change was in 2022, when the percentage dropped to 54% (633) from 66% (678) in the preceding year. Pedestrian injuries showed a 44% reduction: starting at 27% (217) in 2011, and reaching its lowest at 14% (146) in 2021, before showing a slight increase to 15% (173) in 2022. In essence, declining MVC and PED victims concurrent with the expanding MCC populations demonstrated a shift in RTI mechanisms (Figure 3).

### 3.4. RTIs by Season

RTI frequency varied across seasons (Figure 4). Winter, Autumn, Spring, and Summer accounted for 27%, 26%, 24%, and 23% of the total injuries (11,153), respectively (Table 2). Missing data accounted for less than 0.5%. Winter recorded the highest number of injuries in 2014 (27.8%), 2015 (27.6%), 2019 (22.7%), and 2020 (30.5%). Similarly, Autumn was the highest in 2011 (27.6%), 2012 (28.1%), 2013 (28.4%), and 2021 (28.1%). Spring had more injuries compared to other seasons in 2017 (26.8%), 2018 (26.4%), and 2022 (26.7%), while Summer only led during 2016 (27.0%). Winter and Autumn were the most frequent injury seasons (four times each), followed by Spring (three times) and Summer (one time).

### 3.5. Demographic Characteristics of the Injured

The RTI distribution varies across different age groups. The age was recorded for 10,743 persons, making up 96% of the hospitalized RTI victims, while 4% of data were missing. The largest proportion of RTIs was observed among individuals aged 20–29 and 30–39. Specifically, the 20–29 age group accounted for up to 32% of cases, while the 30–39 age group ranged up to 24%. These two groups together represented a substantial portion of RTI cases throughout the period (56%).

Between 2011 and 2018, RTIs among children aged 0–9 remained relatively stable, fluctuating between 6 and 9%, representing 5% of the sum. However, from 2019 onward, this proportion saw a decline, reaching as low as 1% by 2021 and 2022. A similar trend was observed among the 10–19 age group, which initially accounted for 14–15% but dropped to 9% in the later years. This age group accounted for 12% of the total injuries.

Furthermore, the 40–49 age group fluctuated between 12 and 16% over the study period. This group represented 14% of the total, while the group aged 50–59 years comprised 8%. The elderly population (60 years and above) consistently represented a small proportion of RTIs, ranging between 3 and 4% for the 60–69 age group and 0.8–2.3% for those aged 70 and above (Table 1).

In summary, individuals between 20 and 39 years made up the majority of RTI cases, accounting for a significant share across all years. Meanwhile, RTIs among children (0–9 years) and adolescents (10–19 years) declined notably after 2019. The proportion of injuries in older adults remained relatively low and stable over the years.

Among the 11,202 RTI victims, less than one percent had missing data for gender. Of the remaining 11,192 patients, 10,084 (90%) were males while, 1108 (10%) were females. The percentage of injured males started at 90% in 2011, peaked at 92% in 2016 and 2020. On the contrary, the proportion of injured females began at 11% in 2011 and dropped to its lowest at 8% in 2016 and 2020. Of note, among the country populations, males decreased from 74% in 2011 to 72% in 2022. In contrast, females increased from 26% in 2011 to 28% in 2021–2022 [7], as shown in Appendix A.

Between 2011 and 2018, RTIs among children aged 0–9 remained relatively stable, fluctuating between 6 and 9%, representing 5% of the total injuries. However, from 2019 onward, this proportion declined, reaching as low as 1% by 2021 and 2022. A similar trend was observed among the 10–19 age group, which initially accounted for 14–15% of RTIs but dropped to 9% in the later years. This age group accounted for 12% of the total injuries.

Furthermore, the 40–49 age group fluctuated between 12 and 16% over the study period. This group represented 14% of the total, while the group aged 50–59 years comprised 8%. The elderly population (60 years and above) consistently represented a small proportion of RTIs, ranging between 3 and 4% for the 60–69 age group and at 0.8–2.3% for those aged 70 and above (Table 1).

## 4. Discussion

### 4.1. Main Findings

Although the overall death rate decreased, out-of-hospital deaths have risen recently. The severe injuries declined, contrasting the less severe ones. MCC surged and MVC dropped. In addition, pedestrian injuries had a considerable reduction. Injuries were most frequent in the Winter and Autumn seasons, followed by Spring, while they were the least in Summer. Those aged 25–29 represented a high-risk group. In general, males were injured more than females. Lastly, the lowest overall RTI mortality was noted during the COVID-19 pandemic, combined with a peak in bike injuries.

### 4.2. National Progress in Reducing RTIs

The government has undertaken multiple initiatives to enhance road safety. These include a commitment to the global DoARS initiative (2011), implementing the NRSS (2013–2022), establishing the Qatar Road Safety Studies Center, and launching several road safety campaigns [5]. These efforts have been supported by traffic law enforcement. New laws were introduced based on risk assessments like speed limits, drunk and drug driving, motorcycle helmet usage, car seatbelts, and restrictions on mobile phone use while driving [14].

Since its inception in 2007, trauma care has evolved into a mature nationwide system, serving approximately three million people in 2022. This evolution encompasses key milestones such as participation in the international trauma data bank for benchmarking (2011) and Trauma Quality Improvement Program (2013) in addition to being accredited by the Council for Graduate Medical Education (2012), Accreditation by Canada International Care (2014), and The Joint Commission as an academic medical center (2016). Moreover, creating a data-dependent injury prevention program (2012) and trauma research office (2011) have been critical to informing trauma public health decisions [14]. The injury prevention program is involved in a wide range of prevention projects, spanning topics such as pediatric injuries, alcohol screening in trauma patients, and the financial implications of pedestrian injuries, to name a few [4]. These endeavors are geared toward advancing the comprehension of injury epidemiology and protecting vulnerable and high-risk populations through injury prevention.

### 4.3. Decline in RTI-Associated Fatalities

The overall RTI death rate decreased over the study period. In other words, the country met the national set goal by 2019 and sustained the low rate for 2020 and 2021 before a rise in 2022. Injury severity from road crashes decreased as well; there was a shift from very severe to less severe injuries. The following section highlights some key steps taken by the country to overcome RTIs in terms of traffic laws, trauma medical care, and injury prevention.

To provide context for the observed differences in prehospital and in-hospital mortality, it is important to consider the country’s healthcare infrastructure. The public health system is led by Hamad Medical Corporation (HMC), which manages 13 hospitals, including emergency, trauma, and specialized care facilities. Primary healthcare had expanded from 21 to 27 health centers by 2020, while the private sector includes over 190 clinics, 300 healthcare centers, and 5 general hospitals [15]. Qatar’s healthcare workforce includes approximately 7.74 physicians per 10,000 population, compared to 17.2 globally and 11.6 for the Mediterranean region. Additionally, nurses represented 11.87 per 10,000 population, with a hospital bed density of 12 per 10,000 [16]. Emergency medical services are provided by the HMC’s Ambulance Service, which operates over 167 ambulances, 20 rapid response vehicles, and a fleet of helicopters, responding to more than 100,000 calls annually [17].

The overall observed mortality reduction was mainly driven by reduced in-hospital mortality. The proportions of in-hospital deaths (41%) and prehospital deaths (59%) in the total were different from those in the United Arab Emirates, which reported 13.5% in-hospital deaths and 86% prehospital deaths [18]. This can be attributed to the mature trauma system in Qatar [7], as described earlier in the discussion in the Section 4.2. Homayoun et al. also found that the crash location, a counterpart vehicle, street lighting, and a secondhand vehicle were predictors of in-hospital RTI deaths [19], highlighting research opportunities in Qatar. Of note, the lowest overall fatality rate at 5 per 100,000 persons was seen in 2020, coinciding with the imposed COVID-19 restrictions [20]. Likewise, a systematic review by Shaik and Ahmed [21] affirmed that lockdown restrictions reduced RTIs.

The NRSS aimed to reduce the annual number of road fatalities to 6/100,000 individuals by 2022 [6]. Rate comparison is necessary since the populations (denominator) grew by 69% during the study period [7]. For instance, from 2016 to 2021, the death rate remained consistently low at around 5–7 per 100,000 persons, representing a 42–58% reduction compared to 2011. That decline indicates that the RTI mortality reduction target was met. However, there was a slight increase in 2022 (8 deaths/100,000 persons). Furthermore, Consunji et al. reviewed the country’s progress with the DoARS initiative [5]. The authors highlighted variable degrees of compliance with the five-pillar DoARS strategy. For instance, Qatar was found to be fully compliant with Pillars 1 (Road Safety Management) and 2 (Safer Roads) based on the assessment of the DoARS indicators. However, the country only partially met the indicators for Pillars 4 (Safer Road Users) and 5 (Post-crash Response). None of the indicators for Pillar 3 (Safer Vehicles) were met. It is worth mentioning that the average annual number of newly registered vehicles increased from 5255 in 2018 to 7893 by 2022 [22], which may have increased the chances of RTIs due to additional use of the roads. That said, there is a chance to improve RTI mortality and lessen severity by pursuing non-compliant DoARS indicators associated with Safer Vehicles and Safer Road Users.

The increasing trend of prehospital mortality could be associated with crash severity or prehospital care provided at the crash scene. The time between injury occurrence and ambulance arrives at the scene is particularly important, as longer prehospital time has been associated with higher mortality among severely injured patients [23]. Another possible explanation for the recent increase in deaths outside the hospital is a change in ambulance standards of practice implemented in 2019. In that year, the ambulance service updated its criteria for declaring death at the accident scene. As a result, more victims are declared dead at the scene, increasing the number of prehospital deaths and reducing the number of in-hospital ones. Prehospital mortality warrants further research to explore the underlying reasons, focusing on prehospital care, injury severity, RTI mechanism, and use of protective equipment, to develop matching injury prevention programs.

### 4.4. Shift Toward Less Severe Injuries

The country aimed to reduce serious injuries to 15/100,000 people by 2020. “Serious injury is an injury where a person is detained in hospital (e.g., fractures, concussion, internal injuries, crushing, burns”, as defined in the NRSS [6]. Alternatively, this study used the ISS to gauge injury severity, which, in turn, reflects the adequacy of injury prevention effectiveness. Over the study period, a shift from more severe to less severe injuries was proven by reduced populations with a high ISS, corresponding with increasing populations with a low ISS. Lower injury severity may relate to the trauma system improvements described above. Yet, there is a chance to achieve further reduction by introducing new injury prevention measures targeting high-risk groups.

### 4.5. Changing Patterns of Injury Mechanism

Cyclist injuries are uncommon in Qatar [24], similar to the abovementioned findings (Figure 3). Bicycle injuries only spiked during 2021, overlapping with COVID-19 restrictions imposed in the country. The government announced the closure of schools, universities, and all public transport on 10th March 2022 [20]. Multiple studies have shown that there was a rise in bicycle injuries during the pandemic because cycling became more popular as a leisure activity or a fitness regimen, along with a decreased use of public transport for commuting [25,26,27,28].

Furthermore, MVC injuries’ proportion dropped in 2022, corresponding with the increasing MCC category. The growth of MCC injuries can be explained by the increased use of roads. For instance, the number of registered motorcycles rose from 12,542 in 2012 to 16,479 in 2016, recording around a 31% increment [29] on top of the 50% growth in newly registered vehicles between 2018 and 2022 [22]. Motorcycle demand in the country increased by over 55% in 2018 compared to the previous year. This growing demand was due to many businesses, like retailers, restaurants, and fast-food chains, transitioning from cars to motorbikes for service deliveries. This transition aimed to reduce operational and overhead expenses, ensuring these companies remain competitive in the market [30]. Because pedestrians are among the most vulnerable groups to RTIs, the injury prevention program promoted safety education and organized awareness events through partnerships with government bodies and ministries [4] in addition to creating hundreds of pedestrian walkways [31]. The declining trend of pedestrian injuries can be seen as a result of these efforts. To conclude, more injury-preventive activities are needed to target vulnerable groups, especially the growing MCC.

### 4.6. Seasonal Variation in RTIs

Winter and Autumn RTIs were more frequent than other seasons due to weather changes. Moderate temperature promotes outdoor activities during both seasons. On the contrary, elevated temperatures and school shut down lead many citizens to stay indoors or travel abroad for vacation during the summer season [32]. The authors found that these factors led to road user behavior changes, reflected in increased mortality and injury severity among pedestrians and drivers.

This seasonal variation may also reflect shifts in traffic density and patterns, such as increased vehicle use during school months and holiday travel peaks [33]. In Winter and Autumn, the more pleasant weather may encourage walking, biking, and other vulnerable road user activities, increasing exposure to traffic hazards. Moreover, fog and early darkness during cooler months may impair visibility, contributing to increased RTI risks [33]. Understanding these seasonal dynamics can support targeted awareness campaigns and resource allocation. Injury prevention requires exploring the mechanism of injuries and road user behavior across seasons or months of higher injury frequency to develop suitable preventive measures.

### 4.7. Changing Demographics of Injured Individuals

The results presented earlier showed that the age group 25–39 accounted for the majority of the injured (56%), representing an increased RTI risk. That finding corroborated the conclusion of Consunji et al. 2015 [34]. Qatari males aged 20 to 29 years had the highest relative risk of RTI fatality. The results also showed consistently low RTIs among people of 60 years and above (2% on average), compared to the meta-analysis findings (23.4%). The low incidence indicates less vulnerability among this group in Qatar [35]. RTI reduction among kids of nine years or less requires further research to understand the reasons behind the epidemiological change. Nonetheless, it shows that age-specific preventive strategies are needed, particularly among the most affected age groups.

The proportion of males in the country’s population was always larger than that of females [13], which can justify why males consistently represented a larger portion of the injured. Several studies in the Gulf region reported similar results; of two from Saudi Arabia, one (a cross-sectional study) reported 88% of the injured were males [36], while a systematic review reported that young males were injured more than females consistently across most studies [37]. Other research from the Sultanate of Oman also showed the same pattern [38]. Despite a growing female population, males faced higher road injury risks because of increased exposure, occupational driving, or different road behaviors for males, requiring further research.

## 5. Strengths

This study’s main strength lies in its comprehensive, long-term analysis using national data from Qatar, covering a 12-year period. The integration of both in-hospital and prehospital data provides a complete mortality picture, which is uncommon in many trauma studies. The application of a standardized injury severity scoring system (ISS) further enhances the credibility of its trend analysis. This research also succeeds in identifying key shifts in injury mechanisms, seasonal trends, and demographic profiles, helping policymakers target interventions. The alignment of the study period with national and global safety strategies NRSS and DoARS adds value to the evaluation of those initiatives’ outcomes.

## 6. Limitations

This study has several limitations. First, it relies on secondary data sources, which may not encompass all pertinent variables, so there remains a risk of missing variables. The findings are specific to Qatar and may not be generalizable to other countries with different road environments or trauma systems. In addition, the study scope does not provide insights about individual, behavioral, psychosocial, or environmental factors that could influence the causes of road crashes or injury severity. Finally, recent changes in prehospital death declaration protocols (introduced in 2019) may have introduced bias by artificially inflating the number of prehospital fatalities.

## 7. Conclusions and Future Directions

RTIs in Qatar remain a public health challenge despite the notable decline in mortality. The increased MCC and prehospital deaths, as well as changing demographics among RTI victims, require targeting injury prevention programs. To build on these findings, future directions should include strengthening the legal and policy frameworks to address these emerging trends. Integration of artificial intelligence for crash prediction and prevention may also be considered. Continuous monitoring, legal reinforcement, and public health education campaigns, particularly during high-injury seasons, will help sustain and further the achievements in RTI reduction. On a broader scale, Qatar should fully implement all pillars of the DoARS strategy, especially those on vehicle safety and road user behavior. Regular evaluation and adaptation of safety measures aligned with global best practices are crucial for the country’s ongoing road safety success. Enhancing data collection methods in the national registry is necessary to overcome the missing data. Future research should explore the characteristics, mechanisms of injury, and road use behaviors among vulnerable groups such as people aged 25–39 and motorcyclists, using mixed-methods or qualitative approaches. The influence of increasing registered vehicles and the COVID-19 pandemic on road safety can also be studied.

## Figures and Tables

**Figure 1 healthcare-13-01045-f001:**
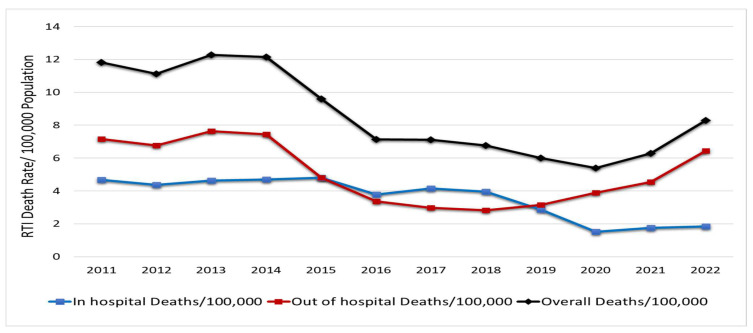
RTI mortalities/100,000 populations in Qatar (2011–2022) [13].

**Figure 2 healthcare-13-01045-f002:**
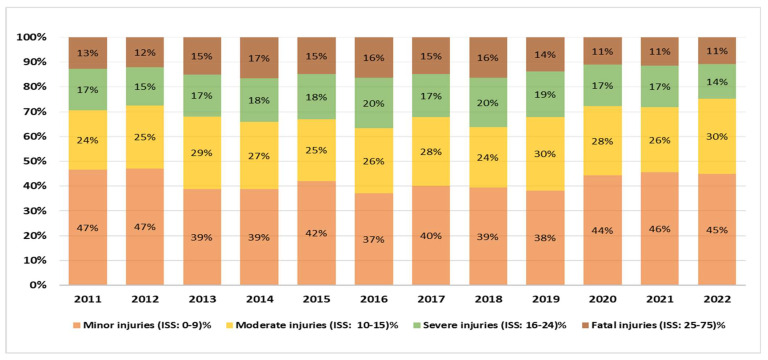
RTI classification by ISS in Qatar (2011–2022).

**Figure 3 healthcare-13-01045-f003:**
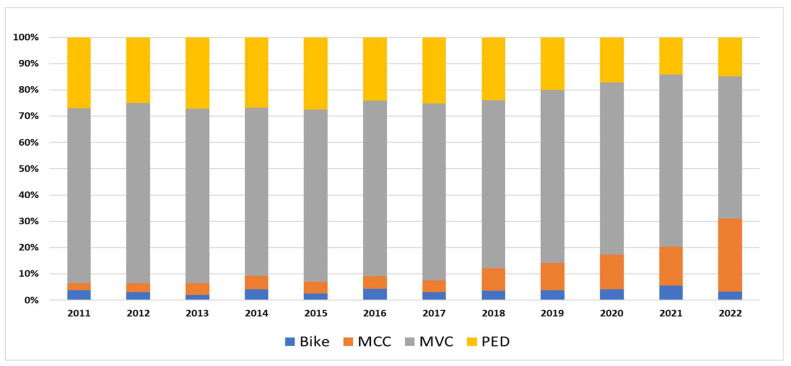
RTI classification by mechanism of injury in Qatar (2011–2022).

**Figure 4 healthcare-13-01045-f004:**
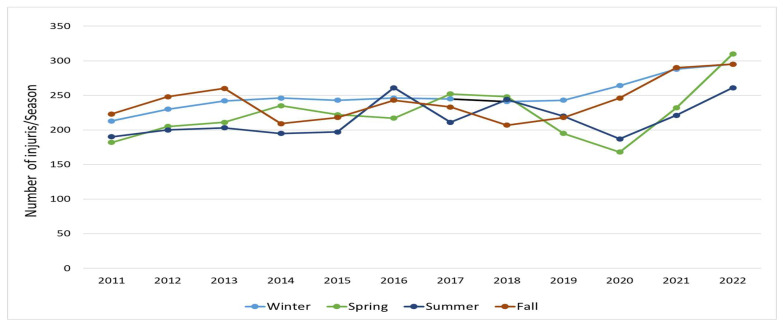
RTI classification by number of injuries/season in Qatar (2011–2022).

**Table 1 healthcare-13-01045-t001:** Demographic characteristics of the trauma patients over the years (*n* = 12,644).

Populations	Characteristics	2011 Number (%)	2012 Number (%)	2013 Number (%)	2014 Number (%)	2015 Number (%)	2016 Number (%)	2017 Number (%)	2018 Number (%)	2019 Number (%)	2020 Number (%)	2021 Number (%)	2022 Number (%)	Total Number (%)
Age in years	0–9	63 (8%)	53 (6%)	66 (7%)	73 (9%)	55 (6%)	54 (6%)	62 (7%)	52 (6%)	30 (3%)	17(2%)	15 (1%)	12 (1%)	552 (5%)
	10–19	104 (14%)	130 (15%)	127 (14%)	110 (13%)	103 (12%)	86 (9%)	120 (13%)	95 (11%)	80 (9%)	79(9%)	112 (11%)	107 (9%)	1253 (12%)
	20–29	258 (34%)	262 (30%)	303 (34%)	242 (29%)	295 (34%)	298 (33%)	267 (30%)	251 (30%)	283 (33%)	263(31%)	326 (33%)	433 (38%)	3481 (32%)
	30–39	159 (21%)	200 (23%)	175 (19%)	201 (24%)	189 (22%)	233 (25%)	227 (25%)	198 (24%)	225 (26%)	257(30%)	268 (27%)	296 (26%)	2628 (24%)
	40–49	100 (13%)	122 (14%)	115 (13%)	105 (13%)	106 (12%)	137 (15%)	114 (13%)	126 (15%)	141 (16%)	119(14%)	164 (16%)	165 (14%)	1514 (14%)
	50–59	55 (7%)	68 (8%)	78 (9%)	61 (7%)	73 (8%)	69 (8%)	65 (7%)	72 (9%)	66 (8%)	82(10%)	72 (7%)	76 (7%)	837 (8%)
	60–69	23 (3%)	24 (3%)	26 (3%)	19 (2%)	28 (3%)	28 (3%)	29 (3%)	28 (3%)	25 (3%)	29(3%)	36 (4%)	41 (4%)	336 (3%)
	70+	6 (0.8%)	10 (1.2%)	9 (1%)	13 (1.6%)	12 (1.4%)	11 (1.2%)	17 (1.9%)	19 (2.3%)	13 (1.5%)	10(1.2%)	9 (0.9%)	13 (1.1%)	142 (1.3%)
Gender	Overall	807	894	926	867	896	961	956	910	895	877	1034	1169	11,192 (100%)
	Males	722 (89.5%)	806 (90.2%)	841 (90.8%)	758 (87.4%)	808 (90.2%)	882 (91.8%)	862 (90.2%)	809 (88.9%)	798 (89.2%)	810 (92.4%)	925 (89.5%)	1063 (90.9%)	10,084 (90.1%)
	Females	85 (10.5%)	88 (9.8%)	85 (9.2%)	109 (12.6%)	88 (9.8%)	79 (8.2%)	94 (9.8%)	101 (11.1%)	97 (10.8%)	67 (7.6%)	109 (10.5%)	106 (9.1%)	1108 (9.9%)

**Table 2 healthcare-13-01045-t002:** Outomes of the trauma patients over the years (*n* = 12,644).

Populations	Characteristics	2011 Number (%)	2012 Number (%)	2013 Number (%)	2014 Number (%)	2015 Number (%)	2016 Number (%)	2017 Number (%)	2018 Number (%)	2019 Number (%)	2020 Number (%)	2021 Number (%)	2022 Number (%)	Total Number
Length of stay	Total in-hospital	6 (3–14)	5 (2–12)	5 (2–13)	5 (2–14)	5 (2–13)	5 (2–14)	4 (2–12)	4 (2–13)	4 (1–12)	4 (2–13)	4 (1–12)	4 (1–13)	5 (1–12
Shock index	ED	0.80 ± 0.46	0.79 ± 0.48	0.76 ± 0.28	0.81 ± 0.28	0.78 ± 0.27	0.79 ± 0.29	0.78 ± 0.27	0.76 ± 0.30	0.78 ± 0.30	0.76 ± 0.38	0.80 ± 1.49	0.74 ± 0.38	0.78 ± 0.58
	prehospital	124	124	153	165	117	88	81	78	88	110	125	189	1442 (58.6)
Mortalities	In-hospital	81	80	93	104	117	99	113	109	80	43	48	54	1021 (41.4%)
Injury severity by ISS *	ISS: 0–9 *	357 (47%)	402 (47%)	347 (39%)	316 (39%)	358 (42%)	350 (37%)	374 (40%)	338 (39%)	334 (38%)	383 (44%)	465 (46%)	521 (45%)	4545 (41.9%)
	ISS: 10–15 *	184 (24%)	217 (25%)	262 (29%)	222 (27%)	213 (25%)	249 (26%)	257 (28%)	210 (24%)	258 (30%)	244 (28%)	268 (26%)	350 (30%)	2934 (27.1%)
	ISS: 16–24 *	128 (17%)	131 (15%)	151 (17%)	143 (18%)	154 (18%)	192 (20%)	162 (17%)	170 (20%)	162 (19%)	144 (17%)	172 (17%)	164 (14%)	1873 (17.3%)
	ISS: 25–75 *	97 (13%)	104 (12%)	135 (15%)	135 (17%)	127 (15%)	155 (16%)	139 (15%)	141 (16%)	120 (14%)	96 (11%)	116 (11%)	125 (11%)	1490 (13.7%)
Mechanism ofinjury	Bicycle	31 (3.8%)	27 (3.0%)	19 (2.1%)	35 (4.0%)	23 (2.6%)	41 (4.3%)	29 (3.0%)	33 (3.6%)	33 (3.7%)	36 (4.1%)	57 (5.5%)	37 (3.2%)	401 (3.6%)
	MCC	21 (2.6%)	30 (3.4%)	41 (4.4%)	45 (5.2%)	39 (4.4%)	47 (4.9%)	43 (4.5%)	77 (8.5%)	94 (10.5%)	116 (13.2%)	153 (14.8%)	326 (27.9%)	1032 (9.2%)
	MVC	538 (66.7%)	613 (68.6%)	614 (66.3%)	555 (64.0%)	587 (65.5%)	642 (66.8%)	643 (67.3%)	582 (64.0%)	589 (65.7%)	575 (65.6%)	678 (65.6%)	633 (54.1%)	7249 (64.8%)
	Pedestrian	217 (26.9%)	224 (25.1%)	252 (27.2%)	232 (26.8%)	247 (27.6%)	231 (24.0%)	241 (25.2%)	218 (24.0%)	180 (20.1%)	150 (17.1%)	146 (14.1%)	173 (14.8%)	2511 (22.4%)
	Winter	213 (26.4%)	230 (26.0%)	242 (26.4%)	246 (27.8%)	243 (27.6%)	246 (25.4%)	245 (26.0%)	241 (25.6%)	243 (22.7%)	264 (30.5%)	288 (27.9%)	295 (25.4%)	2996 (26.9%)
	Spring	182 (22.5%)	205 (23.2%)	211 (23.0%)	235 (26.6%)	222 (25.2%)	217 (22.4%)	252 (26.8%)	248 (26.4%)	195 (22.3%)	168 (19.4%)	232 (22.5%)	310 (26.7%)	2667 (24.0%)
	Summer	190 (23.5%)	200 (22.7%)	203 (22.2%)	195 (22.0%)	197 (22.4%)	261 (27.0%)	211 (22.4%)	244 (26.0%)	220 (25.1%)	187 (21.6%)	221 (21.4%)	261 (22.5%)	2590 (23.2%)
Season of injury	Autumn	223 (27.6%)	248 (28.1%)	260 (28.4%)	209 (23.6%)	218 (24.8%)	243 (25.1%)	233 (24.8%)	207 (22.0%)	218 (24.9%)	246 (28.4%)	290 (28.1%)	295 (25.4%)	2890 (25.9%)
	Overall	768	872	898	826	862	917	897	842	856	854	1002	1142	10,736 (100%)

* ISS: injury severity score; mild injury (0–9), moderate injury (10–15), severe injury (16–24) and fatal injury (25–75).

## Data Availability

All data were presented in this manuscript.

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
