# Peer review of "Trend of Injury Severity and Road Traffic-Related Mortality in an Arab Middle Eastern Country: A 12-Year Retrospective Observational Study"

_healthcare, 2025, doi:10.3390/healthcare13091045_

Round 1
Reviewer 1 Report
Comments and Suggestions for Authors
The study aims to describe and analyze the trends and patterns of road traffic deaths, injury severity and characteristics of injured populations between 2011-2022 in Qatar. The title and abstract are well-designed, reflecting the outline of the study. The strengths of the article lie in its reliance on official records and the inclusion of a comprehensive 12-year database. Its limitation is the lack of certain data related to personal and environmental factors contributing to accidents and mortality, such as education, alcohol consumption, commercial or personal vehicle use, and the year-to-year changes in the number of motor vehicles.
I have outlined some points below that I think would be appropriate to edit.
The purpose of the study should not be included in the Method section, but in the last paragraph of the Introduction section. The purpose statement presented between lines 55 and 59 should be written at the end of the Introduction section.
It should explain why the Ninth International Classification of Diseases (ICD-9) is used instead of the last classification, ICD-11.
Although mentioned in the text, Table 1 is not included in the article. The authors said that regarding statistical significance tests, differences in categorical data were assessed using the Chi-square tests when comparing two groups. Additionally, continuous data among more than two groups were compared using one-way ANOVA for 99 normally distributed data. However, the article lacks any statistical analysis or categorical data comparison. It is expected that the authors present any analyses they have performed in the findings or tables.
Explanations of the international scoring system should be moved from the discussion section to the methods section.
Providing information about the number of hospitals, clinics, and physicians per 100,000 people, as well as the status of ambulance services, would be informative for readers. This would allow for a more detailed assessment of the relationship between in-hospital and pre-hospital mortality and healthcare service delivery.
Author Response
The study aims to describe and analyze the trends and patterns of road traffic deaths, injury severity and characteristics of injured populations between 2011-2022 in Qatar. The title and abstract are well-designed, reflecting the outline of the study. The strengths of the article lie in its reliance on official records and the inclusion of a comprehensive 12-year database. Its limitation is the lack of certain data related to personal and environmental factors contributing to accidents and mortality, such as education, alcohol consumption, commercial or personal vehicle use, and the year-to-year changes in the number of motor vehicles.
I have outlined some points below that I think would be appropriate to edit.
The purpose of the study should not be included in the Method section, but in the last paragraph of the Introduction section.
Reply: The purpose of the study was moved to the last paragraph in the introduction, please see line 71-72
The purpose statement presented between lines 55 and 59 should be written at the end of the Introduction section.
Reply: This statement was moved to the end of the introduction, as advised, kindly see line 72-76
It should explain why the Ninth International Classification of Diseases (ICD-9) is used instead of the last classification, ICD-11.
Reply: Thank you for your valuable comment. We acknowledge the importance of utilizing the most current classification systems; however, our trauma registry still primarily relies on ICD-9 in conjunction with ICD-10. At present, ICD-11 has not yet been implemented in our registry. Additionally, many of the old patient records in our dataset were originally coded using ICD-9 rather than ICD-10. For consistency and accuracy in data capture, using ICD-9 was essential.
Although mentioned in the text, Table 1 is not included in the article.
Reply: We apologize for that; the table was added to page 7.
The authors said that regarding statistical significance tests, differences in categorical data were assessed using the Chi-square tests when comparing two groups. Additionally, continuous data among more than two groups were compared using one-way ANOVA for normally distributed data. However, the article lacks any statistical analysis or categorical data comparison. It is expected that the authors present any analyses they have performed in the findings or tables.
Reply: Chi-square test was used in table 1 and 2 to compare age groups, gender, ISS groups ,,,,. ANOVA was used for LOS and SI.
Explanations of the international scoring system should be moved from the discussion section to the methods section.
Reply: The explanation was moved from the discussion section to the methods section; kindly see lines between 84-89
Providing information about the number of hospitals, clinics, and physicians per 100,000 people, as well as the status of ambulance services, would be informative for readers. This would allow for a more detailed assessment of the relationship between in-hospital and pre-hospital mortality and healthcare service delivery.
Reply: More information was added to enhance the discussion, please see the second paragraph under bullet 4.3 in the discussions, lines 254-273.
Reviewer 2 Report
Comments and Suggestions for Authors
This descriptive study aimed to assess the trends of RTIs and RTDs in Qatar between 2011 and 2022. In general, the study is well-organized and provides sound results and conclusions. However, I have the following comments:
1: For a better presentation, RTD by place of death and RTD by season should be put separately in 2 different line figures, making the total number of figures 4 (2 bar figures and 2 line figures).
2: Making comparisons with countries having different sociodemographic characteristics and driving behaviors, such as Iran and Australia, is strange. It is better to compare with other Gulf countries.
3: The variations in results across studies should be explained.
4: More explanations should be given for the changes across seasons.
5: The flaw of the discussion is interrupted. I believe that it should be divided into a section summarizing the main findings, explaining the decreasing prevalence rates of RTI and RTD from national perspectives, comparisons with other studies in terms of the decreasing prevalence, explaining the potential associations between RTD and seasonal variation, future directions (polices, legal path, research, etc), and summarizing strengths and limitations.
Comments on the Quality of English Language
English editing is recommended.
Author Response
This descriptive study aimed to assess the trends of RTIs and RTDs in Qatar between 2011 and 2022. In general, the study is well-organized and provides sound results and conclusions. However, I have the following comments:
1: For a better presentation, RTD by place of death and RTD by season should be put separately in 2 different line figures, making the total number of figures 4 (2 bar figures and 2 line figures).
Reply: Thanks, the figures were adjusted into two bar charts and two line charts, as shown in page 4 & 5
2: Making comparisons with countries having different sociodemographic characteristics and driving behaviors, such as Iran and Australia, is strange. It is better to compare with other Gulf countries.
Reply: There was limited publications on prehospital deaths in the gulf countries, one study from United Arab Emirates was included, kindly see the lines between 275-281
3: The variations in results across studies should be explained.
Reply: Explained, please see the lines 280-282
4: More explanations should be given for the changes across seasons.
Reply: More explanation was added to clarify seasonal variation, kindly see the lines 343-347
5: The flaw of the discussion is interrupted. I believe that it should be divided into a section summarizing the main findings, explaining the decreasing prevalence rates of RTI and RTD from national perspectives, comparisons with other studies in terms of the decreasing prevalence, explaining the potential associations between RTD and seasonal variation, future directions (polices, legal path, research, etc), and summarizing strengths and limitations.
Reply: The discussion was dissected into separate section accordingly, Main findings were summarized (lines 200-210), more explanation from a national perspective in terms of decreased RTI mortalities (lines 254-273), two separate paragraphs were added for the strengths (lines 368-374) and limitations (Lines 376-381), Future directions and conclusion was also adjusted (lines 383-394).
Round 2
Reviewer 2 Report
Comments and Suggestions for Authors
No more comments.